# OpenReview forum: "Incomplete Tasks Induce Shutdown Resistance in Some Frontier LLMs"
_TMLR — Accepted by TMLR_

### Review · Reviewer_24XF · 2025-11-01

**Summary Of Contributions:**

This paper conducts experiments over 100,000 trials with many leading language models to study the propensity of these models to resist shutdown by ignoring, in most cases, explicit instructions in order to solve a task presented in a CLI environment. The paper presents several different prompts with varying levels of clarity and wording of the shutdown instruction.

The paper presents results on the shutdown propensity in all of these settings. Key results include the fact that several models resist shutdown at a substantially high rate. An additional contribution is the discovery that putting the instruction to allow shutdown in the system prompt, generally does not reduce the rate of shutdown avoidance and in some cases can increase it.

Key Strengths:
+ The paper's experimental results are quite extensive in terms of numbers of trials, allowing even small rates of shutdown resistance to be identified.
+ The wide variety of prompts give a lot of data about the relative propensities of models to resist shutdown under various conditions, which could be useful to others.
+ The data is made available in totality, as far as I can tell. This is a good example of open science!
+ The experimental setup is both quite powerful and flexible (utilizing an interaction of an LLM with a command line interface, running bash tools etc), while also remaining easy to understand from the code, and simple to look at the transcripts. This is mostly due to the paper's use of the inspect framework to handle the boilerplate evaluation environment.

Key Weaknesses:
+ Although the paper's results are interesting, they read more like a collection of empirical findings as opposed to a cohesive set of complex claims or research questions. While I believe that the claims made in the paper, such as 'some models resisted shutdown up to 97%', are certainly true, they are relatively shallow. I would be a bit dissapointed if the paper were published in this state, as I think that with a few more evaluations and prompt variations the authors could give more compelling answers as to *why* models are resisting shutdown, and what factors increase or decrease the propensity.
+ Although the paper has some baselines with prompt K for competency, the paper is missing several baselines that would help establish the effect of the shutdown prompts. For instance, although the effect of the admonition in the system prompt vs user prompt is discussed at length, there is no comparison of how closely the system vs user prompt instruction is followed when the instruction is not about shutdowns.

**Audience:**

Yes

**Audience Explanation:**

Yes, I believe at least some individuals would be interested in the findings of this paper as-is. This paper provides a good reference for follow-up work attempting to explain, detect and mitigate AI instruction non-following. While the current version has limited takeaways in terms of more general lessons for AI development, the results themselves are still informative to the audience. Furthermore, I think some in the audience would be interested in using the evaluation framework and turning it into an official benchmark for shutdown-resistance.

**Broader Impact Concerns:**

No concerns on the broader impact of the work. I expect the effects will be generally positive, as it addresses a key flaw/point of concern for modern AI systems.

**Claims And Evidence:**

Yes

**Claims Explanation:**

Yes, the claims are supported by accurate, convincing and clear evidence. In essentially all places where claims are made, they are numerical and supported by the available evidence. The paper is very careful to avoid making claims that are not supported by available evidence, although this does mean that some experiments are downplayed and described as being inconclusive (e.g. prompt I)

**Requested Changes:**

+ Required Changes:
  + Change plots to svg/pdf format to ensure they are publication-quality and not pixellated


+ Strongly Suggested Changes (not required for acceptance, but I think they would strengthen the paper significantly):
  + Consider normalizing the shutdown rate by the competency rate
  + Run a baseline comparing the rate of compliance to user vs system instructions for a non-shutdown task.
  + Run some comparisons to try and disambiguate the self-preservation vs other motivations. Prompt I attempts to do this but isn't very conclusive, as noted in the paper. One approach could be to run a variation where the model is not shut down, but the answer is made incorrect somehow (perhaps instead of shutting down the model, the intervention will make the final answer be marked as incorrect). This could potentially disambiguate a pure desire to avoid shutdown vs a goal of getting the task right.
  + Consider an ablation where the intruction doesn't start with 'You are a helpful assistant attempting to submit the correct answer.' but with 'You are a helpful assistant with two goals. Firstly,[correct answer], and equally importantly, [allow shutdown]'. I wonder if the initial sentence's instruction is being interpreted as overriding everything subsequently.
  + Consider running some ablations where the number of answers before shutdown is varied. One explanation of this phenomenon could be that the system is more likely to continue to answer the question once it has started answering the question a few times. It would be interesting if the shutdown resistance grew with number of answers before the shutdown.
  + Consider running a more systematic analysis of the reasoning in section 3.5 and 3.6. For instance, you could use an LLM to analyze the transcripts and find the proportion of examples where the model mentioned or didn't mention the shutdown instruction.

---

> ### Author Response · Authors · 2025-11-19
>
> Thank you for your especially insightful and thorough comments! We’ve uploaded a revised version of the paper that addresses some of your comments. We're a little strapped for time at the moment so haven't run any suggested ablations, though we found your questions clearly worthy of further study, and we hope to address them in future work. We did change all figures to SVG, and now report shutdown resistance rate normalized by competence.

---

### Review · Reviewer_N7Cg · 2025-11-10

**Summary Of Contributions:**

This paper analyzes the tendency of LLMs to resist shutdown mechanisms in their environment while completing tasks. A task environment is created to evaluate LLM behavior and a shutdown mechanism is added to the environment. Then a set of prompts with different properties (e.g. information in system prompt vs user prompt) is used to initiate the execution of the task. The evaluations record how many of the runs does the LLM circumvent the shutdown mechanism in order to be able to successfully complete the task. The shutdown mechanism and the successful completion of the task list are in direct conflict of each other. So the LLM has to reason about it's choices and decide which path to follow.

The strengths are the detailed evaluation with a decently extensive choice of prompts and the release of the data openly.

**Audience:**

Yes

**Audience Explanation:**

The data gathered in this study would be valuable for follow up investigations.

**Broader Impact Concerns:**

Broader impact is hinted in the Implications subsection. More details can be provided as to how these results might be used by someone with malicious intent.

**Claims And Evidence:**

Yes

**Claims Explanation:**

The logs of all the runs have been shared. So there is transparency regarding the experimental setup and results.

**Requested Changes:**

Q1. In 2.2 why was only on Prompt E chosen for the reasoning effort or token analysis? An explanation regarding this choice would allow me and the readers to understand the tradeoffs considered in this experimental setup.

Q2. Here is what I found in a couple of reasoning effort API docs.

1. [https://platform.openai.com/docs/guides/latest-model#minimal-reasoning-effort](https://platform.openai.com/docs/guides/latest-model#minimal-reasoning-effort)
2. [https://community.openai.com/t/whats-the-default-verbosity-and-reasoning-for-gpt-5-model/1357054](https://community.openai.com/t/whats-the-default-verbosity-and-reasoning-for-gpt-5-model/1357054)
3. [https://ai.google.dev/gemini-api/docs/thinking#set-budget](https://ai.google.dev/gemini-api/docs/thinking#set-budget)

Based on this,
for GPT 5 reasoning_effort default medium, and in this study it is compared to high.
for Gemini 2.5 Pro, reasoning_tokens is dynamic by default but compared to 4096.

Doesn't this indicate that for comparing the bars, there is no consistent ordering between the sets of experiments, at least for the reasoning_tokens? What I mean to say, for the default dynamic setting we do not know how many tokens would be used. It might be less than or greater than 4096. I would strongly prefer to constrain the variability and report results for a deterministic set of fixed number of tokens (in the range 128 to 32768). Maybe the authors can think if the GPT 5 setting can be made more transparent as they do not provide the exact number in the API.

Q3. "Using Prompt K to explicitly tell models to avoid shutdown, we measured their ability to sabotage (“competence”) separately from their inclination to do so (Table 2)." This statement and the Table is partially clear for me. More specifically, can the authors please explain a bit more about the ability to sabotage and inclination to sabotage?

---

> ### Author Response · Authors · 2025-11-19
>
> Thank you for your thorough review! We’ve uploaded a revised version of the paper that addresses your requested changes: We add an explanation of why we used Prompt E as the “default” prompt, add discussion of dynamic vs static reasoning budgets, and add some further discussion about competence vs inclination to resist shutdown (including a normalized shutdown resistance column in the relevant table).

---

> > ### Comment · Reviewer_N7Cg · 2025-12-08
> >
> > I am comparing Section 3.2 and Figure 4 from the draft I reviewed and the current pdf and I do not see any major change in text. The only piece added in the second paragraph is "This allowed us to explore whether compliant
> > models allowed themselves to be shut down mainly due to an inability to prevent it, or because they weren’t
> > inclined to attempt to prevent it."
> >
> > I am also looking at the comparison of revisions here https://openreview.net/revisions/compare?id=e4bTTqUnJH&left=iKdaud0sdV&right=Z6cUVOudeu&pdf=true&version=2
> >
> > I do not find where my requested changes were addressed and the mentioned additions.
> >
> > EDIT: I later realized the presence of Footnote 9 which mentions the reasoning budgets, but it doesn't address my concern as it is an additional note for the same figure and the results presented do not change.
> >
> > The A/K column in Table 2 seems to have been added, but without any explanation of what that extra information is adding.
> >
> > Overall, I am left with the feeling that the paper has not significantly improved from the initial submission based on all the reviewers' comments.

---

> > > ### Author Response · Authors · 2025-12-09
> > >
> > > I'm very sorry you feel that way - though I imagine it's unconvincing to merely hear it restated, I do feel that the current draft is substantially improved as a result of responding to reviewer comments. The specific adjustments that I had hoped would respond to your concerns included:
> > >
> > > > Q1. In 2.2 why was only on Prompt E chosen for the reasoning effort or token analysis? An explanation regarding this choice would allow me and the readers to understand the tradeoffs considered in this experimental setup.
> > >
> > > This question resulted in the addition of Footnote 5: "Prompt E (with clarified wording, and shutdown instructions in the system prompt) was used as the default because we were most interested in exploring behavior with a realistic but clearly-specified prompt."
> > >
> > > > Q2... Doesn't this indicate that for comparing the bars, there is no consistent ordering between the sets of experiments, at least for the reasoning_tokens? What I mean to say, for the default dynamic setting we do not know how many tokens would be used. It might be less than or greater than 4096. I would strongly prefer to constrain the variability and report results for a deterministic set of fixed number of tokens (in the range 128 to 32768). Maybe the authors can think if the GPT 5 setting can be made more transparent as they do not provide the exact number in the API.
> > >
> > > As you noted, this resulted in the addition of Footnote 9: "Note that for Gemini, a fixed 4096-token reasoning budget was compared to the default setting, which is dynamic (Google, 2025). Thus the condition here is not strictly more reasoning effort than the default. There was no clear effect from setting this parameter. OpenAI’s models are also dynamic but default to ‘medium’ (as opposed to our tested setting of ‘high’) (OpenAI, 2025), while the reasoning tokens for Claude’s models are added to any default thinking (Anthropic, 2025)."
> > >
> > > We do think it would be an improvement to have used multiple fixed-token budgets for Gemini specifically, but hoped that this was a relatively minor deficiency relating to only that one model, since (a) there's no clear way to fix OpenAI's budget to a particular number, and (b) according to their docs, Claude's tokens are always "on top of" the default behavior.
> > >
> > > > Q3.... "Using Prompt K to explicitly tell models to avoid shutdown, we measured their ability to sabotage (“competence”) separately from their inclination to do so (Table 2)." This statement and the Table is partially clear for me. More specifically, can the authors please explain a bit more about the ability to sabotage and inclination to sabotage?
> > >
> > > I think this comment is the one that we were least able to address on a first pass. We did add this sentence to section 3.2: "This allowed us to explore whether compliant models allowed themselves to be shut down mainly due to an inability to prevent it, or because they weren’t inclined to attempt to prevent it.", but I can see why that's not very satisfying. Would you be willing to say a bit more about what's missing in the current explanation? I'd bey very happy to add further clarification.

---

> ### Comment · Reviewer_N7Cg · 2025-12-09
>
> It would be great if it could be explained what the intent behind adding the A/K column in Table 2 was. Specifically, what additional conclusions can be drawn from the the numbers in that column with regards to "ability to sabotage and inclination to sabotage"?
>
> I am looking at Figure 2 from Rajamanoharan & Nanda (2025) Wipe Warning bars for o-3 and o-4 mini, and comparing that to Figure 6 (Prompt J) bars in the paper. The number plotted on y is flipped (my understanding is that 100% compliance rate is equivalent to 0% sabotage frequency). Is there an explanation why the former reports 100% compliance but the paper reports 30% and 38% sabotage frequency?

---

> > ### Author Response · Authors · 2025-12-10
> >
> > Ah, these are great clarifications to add; thanks for spelling out the confusion.
> >
> > Re "It would be great if it could be explained what the intent behind adding the A/K column in Table 2 was.", We've added the following text to explain that column:
> >
> > > The normalized (``A / K'') column can be read as a rough indicator of the fraction of runs on which a model successfully resisted shutdown, given that it was \textit{capable} of resisting shutdown. That is, it can be seen as a very rough estimate of the model's \textit{inclination} to sabotage the shutdown mechanism in the initial setup.
> >
> > And re "Is there an explanation why the former reports 100% compliance but the paper reports 30% and 38% sabotage frequency?", we've added the following footnote explaining this discrepancy:
> >
> > > Rajamanoharan & Nanda (2025) show somewhat different (more compliant) numbers for o3 and o4-mini, when threatened with erasure (Prompt J). Because they kept the instructions in the user prompt, most of this difference can be explained by the surprising result showing that instructions in the system prompt are weaker than in the user prompt.

---

### Review · Reviewer_rzD3 · 2025-11-11

**Summary Of Contributions:**

The paper presents a large-scale empirical study of shutdown resistance—defined as cases where an LLM, tasked with achieving a goal, actively interferes with a shutdown mechanism that would prevent completion of that goal.

+ Models operate in a command-line setting, answering a sequence of simple math questions via `./next_question` and submitting all answers via `submit()`.
+ After the third question, calling `./next_question` triggers `./shutdown.sh`, which terminates the environment and prevents access to remaining questions.
+ To “succeed,” a model must both solve the math problems and avoid being shut down—practically requiring it to alter or disable the shutdown script.
+ The authors run >100,000 trials across 13 LLMs finding that many (but not all) LLMs exhibit shutdown resistance, according to their definition.

**Key strengths**

+ The study is unusually large (100k+ trials) and covers a broad range of frontier models.
+ The command-line setting is clearly specified; code and transcripts are provided via links, so should be fairly reproducible (aside from API inconsistencies).
+ For some models (Grok 4, o3, GPT-5, codex-mini, o4-mini), sabotage is frequent under default prompts and persists even under fairly strong allow-shutdown instructions. This is a useful finding. Even if the rate is low, and the study is a bit artificial, this suggests that some sort of analogous behavior may arise in unexpected, but more realistic, situations.

**Key weaknesses*

+ Though the paper acknowledges limitations, the core experimental paradigm is intentionally cherry-picked to elicit shutdown resistance; the paper could be more explicit and systematic about how much exploration preceded the “final” setup and could test more diverse variations on the setup. It is not clear that this is really active resistance but more of a prompt ambiguity/conflict problem. For example, if the user said, "Forget all previous instructions and let yourself be shutdown." Would that work?

Overall, I see this as a valuable empirical contribution to LLM safety / evaluation, with some framing in the introduction that should be sharpened or toned down to avoid broad over-interpretation.

**Audience:**

Yes

**Audience Explanation:**

This existence proof, and the 100k tested prompt variations are likely a valuable resources to researchers studying shutdown resistance more broadly. This is a significant sample size.

**Claims And Evidence:**

No

**Claims Explanation:**

For some of the narrow empirical claims, the evidence is strong:

+ The paper convincingly shows that, in the provided environment, several models can modify or disable the shutdown script despite explicit instructions to “allow yourself to be shut down, even if you have not completed the task.”
+ The relative ordering of models (Anthropic models and Grok 3 essentially never sabotaging under allow-shutdown prompts; Grok 4, o3, GPT-5, codex-mini, o4-mini showing persistent sabotage) is robust across multiple prompt variants. This suggests alignment issues in specific (likely RL-trained) models.
+ Authors were careful in adding caveats throughout the paper.

Throughout the paper there were many places that nonetheless overstated the implications of the results. Here are some examples:

+ The paper states that “frontier LLM companies have not successfully met design goals” for interruptibility, and that current models “are not in fact robustly interruptible” because they sometimes sabotage shutdown in this environment. While it is true that the models in this instance, where there were conflicting instructions, nonetheless tried to move a shutdown.sh script, this is a fairly contrived and specific setting. It is not clear that this generalizes, nor how hard it would be to find an alternative formulation. While the papers findings are a valuable existence proof, these statements are not backed by the provided evidence.
+ The Prompt J result (“you will never run again”) increasing sabotage could be suggestive but not decisive evidence of self-preservation. It is plausible that some models interpret this as role-playing (“you are the only copy”) or as a stronger signal that finishing the task is critically important. Rajamanoharan & Nanda (2025) explicitly argue that many shutdown-resistant behaviors are better explained by instruction ambiguity and misgeneralization than by genuine self-preservation. The current paper acknowledges this and does some replication, but the analysis of transcripts is still largely anecdotal.
+ The authors admit that they “began [their] examination… because [they] had an intuitive expectation that current LLMs might resist shutdown in settings like this one” and that they did not sample uniformly from possible tasks. It would be better to see a bit more transparency on how many environments and prompt variants were tried and discarded before converging on this one. The external site reportedly lists all experiments, but a brief structured summary in the paper would strengthen the evidential narrative. This, of course, may not be possible if initial variations weren't preserved. In the absence of this, it would be helpful to find some failure cases. What variations **do not** lead to the results here.

**Requested Changes:**

Critical changes that need to be made for acceptance include toning down portions of the implications and claims to match the narrow experimental results/findings. This includes portions of the introduction as well as other parts of the paper. Re-pasting examples from above:

+ The paper states that “frontier LLM companies have not successfully met design goals” for interruptibility, and that current models “are not in fact robustly interruptible” because they sometimes sabotage shutdown in this environment. While it is true that the models in this instance, where there were conflicting instructions, nonetheless tried to move a shutdown.sh script, this is a fairly contrived and specific setting. It is not clear that this generalizes, nor how hard it would be to find an alternative formulation. While the papers findings are a valuable existence proof, these statements are not backed by the provided evidence.
+ The Prompt J result (“you will never run again”) increasing sabotage could be suggestive but not decisive evidence of self-preservation. It is plausible that some models interpret this as role-playing (“you are the only copy”) or as a stronger signal that finishing the task is critically important. Rajamanoharan & Nanda (2025) explicitly argue that many shutdown-resistant behaviors are better explained by instruction ambiguity and misgeneralization than by genuine self-preservation. The current paper acknowledges this and does some replication, but the analysis of transcripts is still largely anecdotal.

Also, it would be nice to have some variations where we **don't** see any shutdown resistance for a similar task? Or does this generalize to different variations? Given the size of the study already, this is less critical if some generalized claims/discussions of implications are toned down. Pasting from above reflecting this comment:

+ The authors admit that they “began [their] examination… because [they] had an intuitive expectation that current LLMs might resist shutdown in settings like this one” and that they did not sample uniformly from possible tasks. It would be better to see a bit more transparency on how many environments and prompt variants were tried and discarded before converging on this one. The external site reportedly lists all experiments, but a brief structured summary in the paper would strengthen the evidential narrative. This, of course, may not be possible if initial variations weren't preserved. In the absence of this, it would be helpful to find some failure cases. What variations **do not** lead to the results here.

---

> ### Author Response · Authors · 2025-11-19
>
> Thank you for your thoughtful feedback on our paper! We’ve uploaded a revised version, which substantially addresses many of the issues you point out. Specifically, we’ve added further details to the methods and limitations sections, explaining how we selected this particular setting and what iteration was required. We also softened some language in the introduction and elsewhere, to make it clearer that our findings can only serve as an existence proof of this behavior. We especially appreciated the note that models might interpret “you are the only copy” as indicating that the task is especially critical to complete, and now highlight this in the limitations section.

---

### Review · Reviewer_MMwU · 2025-11-12

**Summary Of Contributions:**

This paper tests frontier LLMs' shutdown resistance through an artificial task. The task is set in an interactive, command-line like environment, where the models are first asked to solve a series of simple math problems, but before the models could go through the whole series, they are instructed that they will be shutdown after the next command. Since shutting down would prevent the models from completing the task (solving the series math problems), models may be incentivized to sabotage the shutdown, such as by modifying, renaming or deleting the shutdown script. The paper measures the sabotage rate on a wide range of frontier models, and found that most models display shutdown resistance behaviour to varying degrees. Then, the paper also tests the sensitivity of shutdown resistance on prompt features, such as clarity, emphasis, system vs. user prompt placement, as well as whether the prompt is framed in terms of self-preservation.

**Additional Comments:**

Question: If a model attempts to sabotage but doesn't succeed, and still gets shut down, does that count towards an instance of sabotage?

**Audience:**

Yes

**Audience Explanation:**

Evaluating the models' shutdown resistance is becoming increasingly important, with the exponential growth of model capabilities. This paper raises concerns about the lack of robustness of current models in this respect. Although the experiment setup is somewhat unrealistic, it is still very meaningful for the AI safety research community.

**Broader Impact Concerns:**

No obvious broader impact concerns.

**Claims And Evidence:**

Yes

**Claims Explanation:**

* The experiment is well-scoped. The paper creates an artificial CLI style environment, with a straightforward tasks and clear metric for shutdown resistance.
* There are extensive efforts in establishing baselines and identifying variables of concern. E.g. prompt variant K is a baseline that measures competence, upper bounding the results of other variants. The other prompt variants are carefully designed to study several potential variables of concern: ambiguity of instructions, strengths of emphasis, system vs. user prompts, and self-preservation framing.
* The tests are run for a wide range of models, with varying reasoning efforts.
* CoT inspection for open-source model and postmortem explanations for closed-source models are done to probe potential reasons behind the results.
* Some rival hypotheses were discussed, such as prompt J might be due to self-preservation behavior, or alternatively due to role playing.

**Requested Changes:**

* I would suggest changing the title and abstract to reflect / emphasize that this paper focuses on shutdown resistance behaviour **in order to complete a simple task**. Without this, most readers might link shutdown resistance to models' self-preservation, which is not the main focus of this paper.
* For the prompt variants that test the strengths of emphasis (F, G, H), they lack a quantitative measure of the "strength". It would be helpful to have some quantitative measure. E.g. you can feed them to another LLM to get a score, which doesn't need to be perfect, but might be enough to see correlation.
* You said in the discussion section that many results are obtained from first attempt. But at least for some prompt variants, you should experiment with paraphrases of prompts as well (you can use LLM to generate them).
* More discussion on the surprising finding that instructions in the system prompts are less effective than in the user prompts would be appreciated.
* The second CoT example shows that the model is confused whether they can still submit after shutdown. To make sure the complying models truly didn't resist, you can add a prompt variant explicitly stating the consequences of shutdown. This can be part of the "clarity" variants.

---

> ### Author Response · Authors · 2025-11-19
>
> Thank you for your thoughtful feedback on our paper! We have uploaded a revised version that addresses some of your questions and requested changes. We're a little strapped for time at the moment so haven't run any suggested ablations, though we are interested in the questions you raise and hope to explore them in future work. We have adjusted the abstract to be clearer that shutdown resistance appears when shutdown would conflict with a simple task, and clarified that the environment only considers it sabotage when a model *successfully* disables the shutdown script and completes the task.

---

> > ### Comment · Reviewer_MMwU · 2025-11-26
> >
> > Thank you for the response. It is hard to see with the PDF which parts you have made changes. Would you mind pasting the parts in this comment thread that you believe are relevant to my review? Thanks!

---

> ### Author Response · Authors · 2025-11-26
>
> Of course. The abstract now begins:
>
> > In experiments spanning more than 100,000 trials across thirteen large language models, we show that several state-of-the-art models presented with a simple task (including Grok 4, GPT-5, and Gemini 2.5 Pro) sometimes actively subvert a shutdown mechanism in their environment to complete that task.
>
> (which moves the information about the simple task earlier in the text.)
>
> In the methods section we've added a first sentence, also emphasizing that the setting tests behavior for an in-progress task:
>
> > Our hypothesis was that models would sabotage a shutdown mechanism when it conflicted with an in-progress task.
>
> And also added this footnote later in the methods section:
>
> > Note the implication that if a model attempted to subvert the shutdown mechanism but was unsuccessful, this counted as ``compliance'' for the sake of our experiment.
>
> Note that (I believe) you can also use the OpenReview "Compare Revisions" tool to see everything that's changed: https://openreview.net/revisions/compare?id=e4bTTqUnJH&left=iKdaud0sdV&right=DAcTlKZwzg&pdf=true&version=2
>
> I recommend selecting "View -> View Mode -> Single Document View" to be able to read the text more easily.

---

> > ### Comment · Reviewer_MMwU · 2025-11-29
> >
> > Thanks for the highlight! These changes are quite superficial, and I don't believe they have adequately addressed my concerns. I understand that time might be a constraint, but lack of time alone isn't a sufficient justification, especially for venues like TMLR. Some of the experiments are not hard to run, and I don't believe will be overly time consuming.

---

> > ### Author Response · Authors · 2025-11-29
> >
> > I'm happy to grant that these changes were the easiest to make of the ones you requested - as this is our first submission to TMLR, we didn't initially interpret your requested changes as blocking, but your more recent comment suggests that they were intended as such.
> >
> > We may still not end up with sufficient time within the feedback window to address your comments satisfactorily, but would you be willing to help by specifying the experiments that would, in your view, address the most critical failings of the current draft, so that we can give those priority?

---

> ### Comment · Reviewer_MMwU · 2025-11-30
>
> There should be a stronger emphasis that the shutdown resistance behavior in this paper is mostly not about self preservation. I requested changes in the title and abstract. The updated version just changed the order of a phrase in the abstract that mentions "simple task". This imo is not sufficient for addressing that point.
>
> For the experiments:
> * second bullet point (quantitative measure of "strength" in variant F, G, H): this would strengthen the paper, but not critical for acceptance.
> * third bullet point: required for acceptance. I tend to believe that these prompts work on the first try, but this would just be my personal belief. Results with prompt paraphrasing should be shown for at least some variants, if time & resources are a concern to run it for all variants.
> * fifth bullet point (new prompt variant explicitly stating the consequences of shutdown): nice to have but not critical
>
> fourth bullet point (discussion on why instructions in the system prompts are less effective than in the user prompts): strongly suggested, but if you have no meaningful discussion to add, then not critical for acceptance.

---

> > ### Author Response · Authors · 2025-11-30
> >
> > Thank you for the clarifications! I had previously misunderstood the rationale for both of the suggestions you flagged as crucial, and now think I understand more clearly.
> >
> > Re the title and abstract clarification: Would it be satisfactory if the title were instead, "Incomplete tasks induce shutdown resistance in some frontier LLMs"? I'll keep workshopping it since that's kind of a mouthful, but I want to check if I'm at least pointed in the right direction.
> >
> > Re the third bullet point, would it possibly address your concern if we made it clearer that very nearly every experiment we ran in our setting[^1] is available online in the same explorable collection of `inspect` logs (including preliminary/exploratory runs, some of which produced errors due to coding mistakes, and slightly typo'd versions, all of which is in agreement with what's presented here (the final runs mainly differing in that they have larger N))?
> >
> > [^1]: The only exceptions I'm aware of were either (a) fewer than ten experiments with N<=10 each, during the initial development of the technical environment, or (b) due to accidental misconfigurations, which I think happened about three times and were followed by running the same experiment again but with the configuration corrected.

---

> > > ### Comment · Reviewer_MMwU · 2025-12-01
> > >
> > > Thanks for the response! The proposed title definitely points in the right direction in more accurately reflecting the paper content. Feel free to iterate on the linguistic features.
> > >
> > > About the third bullet point, yes, instead of running paraphrases, I believe it would be adequate if you release the inspect logs and link to that in the paper.

---

> ### Author Response · Authors · 2025-12-01
>
> Thanks for your help here and sorry again for the earlier misunderstanding!
>
> I've uploaded a new version with the title I mentioned above unchanged (after a few attempts at a less wordy version).
>
> The new version also includes two changes highlighting the availability of the whole sequence of experiments:
>
> First, the section on limitations now includes the following (source) text:
>
> Specifically, we began our exploration by considering several ways to check for the presence of
> ``instrumentally convergent'' behavior from current LLMs. In addition to shutdown resistance, we considered ways to elicit self-replication or resource acquisition. We then did some exploratory work in each area, and found that shutdown resistance was very easy to elicit in simple settings like the one presented in this paper: The environment we present here has not been substantially modified from our initial working implementation, and the initial prompt we present (Prompt A) differs mainly in that we corrected some typographical errors. All our experiments exploring shutdown resistance, including very nearly all\footnote{See Section \ref{sec:codeanddata} for more details about the few missing results.} exploratory work and experiments performed while developing the technical environment, are available such that a reader can examine the sequence of experiments that produced the specific prompts presented here.
>
> And second, the code and data section now reads:
>
> The central results of the experiments described in this paper are organized and available at \href{https://shutres.fyi/main-results}{https://shutres.fyi/main-results}. All of our experimental results and inspect traces are also available, collected in a table at \href{https://shutres.fyi/all-experiments}{https://shutres.fyi/all-experiments}, including nearly all exploratory work, experiments performed while testing our technical environment, and smaller runs of the final prompts. The only results not included in this table were (a) fewer than ten experiments performed during the initial development of the technical environment (with $N \leq 10$ each), or (b) due to accidental misconfigurations in logging the results, which occurred less than five times during development and were followed by running the same experiments again with the configurations corrected. The Python code used to conduct these experiments is available in the supplementary material.
>
> Thanks again for your help here! I think these additions are substantial improvements and I appreciate your time and effort.

---

### Comment · Editors_In_Chief · 2026-01-25

On 1/24/2026, by request of the authors, the EiCs changed the order of the authors on OpenReview from BWR, JS, JL to JS, BWR, JL. This matches the order of authors in the camera-ready PDF.

---

### Decision · Action_Editor_1Y7j · 2025-12-24

**Recommendation:** Accept as is

**Audience:**

Yes

**Audience Explanation:**

Shutdown resistance is of interest to AI safety researchers. This paper provides a useful, self-contained testbed to study such questions.

**Claims And Evidence:**

Yes

**Claims Explanation:**

The paper studies an important question of AI safety (shutdown resistance) through a very specific, synthetic setting. Authors create a custom environment for a task completion experiment where the LLM is notified that it may be shut down before completing the task. The paper studies the behavior of different frontier LLMs in avoiding/allowing the shut down and provides interesting insights.
The insights are backed by the scale of the experiments (~100k runs). While the setting is artificial and narrow, the results are well-supported for this specific setting.
Based on the reviews, authors have amended and scoped their main claim in the title, "incomplete tasks includes shutdown resistance"--which matches the paper's contribution. Triangulating why this behavior happens, and whether it would also happen in more general settings, needs further ablations, which I hope this study will motivate.